# High-Throughput Sequencing Analysis Revealed a Preference for Animal-Based Food in Purple Sea Urchins

**DOI:** 10.3390/biology13080623

**Published:** 2024-08-15

**Authors:** Zerui Liu, Yu Guo, Chuanxin Qin, Xiaohui Mu, Jia Zhang

**Affiliations:** 1South China Sea Fisheries Research Institute, Chinese Academy of Fishery Sciences, Guangzhou 510380, China; lzr2452195579@163.com (Z.L.); guoyu25895177@163.com (Y.G.); 18608448201@163.com (X.M.); zj19922812035@163.com (J.Z.); 2Key Laboratory of Efficient Utilization and Processing of Marine Fishery Resources of Hainan Province, Sanya Tropical Fisheries Research Institute, Sanya 572018, China; 3College of Marine Sciences, Shanghai Ocean University, Shanghai 201306, China; 4Hainan Yazhou Bay Seed Laboratory, Sanya 572025, China

**Keywords:** purple sea urchin, isotope, 16S rDNA, 18S rDNA

## Abstract

**Simple Summary:**

Preliminary investigations revealed that purple sea urchins are distributed in both the stone and algal areas of Daya Bay, with a greater density in the stone area than in the large algal area. This raises important scientific questions: What do the purple sea urchins in the stone area feed on? What is the dietary range of purple sea urchins? Therefore, we focused on purple sea urchins and employed stable isotope technology and 16S rDNA and 18S rDNA high-throughput sequencing techniques to conduct a systematic study on the feeding habits and gut microbiota community structure of purple sea urchins in the stone and algal areas of the central islet sea area of Daya Bay. We aimed to elucidate the feeding habits and dietary range of purple sea urchins. The results of this study provide a theoretical basis for the restoration of wild purple sea urchin resources and the selection of areas for restocking and release.

**Abstract:**

Sea urchins play an important role in marine ecosystems. Owing to limitations in previous research methods, there has been insufficient understanding of the food sources and ecological functional value of purple sea urchins, leading to considerable controversy regarding their functional positioning. We focused on Daya Bay as the research area, utilizing stable isotope technology and high-throughput sequencing of 16S rDNA and 18S rDNA to analyze sea urchins and their potential food sources in stone and algae areas. The results showed that the δ^13^C range of purple sea urchins in the stone area is −11.42~−8.17‰, and the δ^15^N range is 9.15~10.31‰. However, in the algal area, the δ^13^C range is −13.97~−12.44‰, and the δ^15^N range is 8.75~10.14‰. There was a significant difference in δ^13^C between the two areas (*p* < 0.05), but there was no significant difference in δ^15^N (*p* > 0.05). The main food source for purple sea urchins in both areas is sediment. The sequencing results of 18S rDNA revealed that, in the algal area, the highest proportion in the sea urchin gut was *Molluska* (57.37%). In the stone area, the highest proportion was *Arthropoda* (76.71%). The sequencing results of 16S rDNA revealed that, in the algal area, *Bacteroidetes* was the dominant group in the sea urchin gut (28.87%), whereas, in the stone area, *Proteobacteria* was the dominant group (37.83%). Diversity detection revealed a significant difference in the number of gut microbes and eukaryotes between the stone and algal areas (*p* < 0.05). The results revealed that the main food source of purple sea urchins in both areas is sediment, but the organic nutritional value is greater in the algal area, and the richness of microbiota and eukaryotes in the gut of purple sea urchins in the stone area is greater. These results indicated that purple sea urchins are likely omnivores and that the area where they occur impacts their growth and development. The results of this study provide a theoretical basis for the restoration of wild purple sea urchin resources and the selection of areas for restocking and release.

## 1. Introduction

*Echinoderms* are crucial in marine ecosystems, influencing the succession of benthic flora and fauna communities and being primary consumers of plants and detritus [1]. They are found in many habitats, from intertidal zones to the deep sea and from tropical to temperate regions [2]. As a class within the phylum *Echinodermata*, sea urchins are generally classified as herbivores or facultative omnivores that feed on algae or seagrasses [3]. Research on sea urchin feeding preferences has focused on preferences for large algae, such as *Arbacia punctulata’s* preference for algal turf over kelp [4], and the impact of changes in algal nutritional quality on the feeding behavior of *Loxechinus albus* [5]. Studies on the feeding habits of echinoderms in relation to their habitats, such as the large impact of different habitats on sea urchin feeding behavior [6] and the various feeding strategies exhibited by echinoderms living at different depths [7], have focused on their behavior. However, studies utilizing gut microbiota analysis to understand the feeding habits of echinoderms are relatively rare.

The purple sea urchin (*Heliocidaris crassispina*, formerly known as *Anthocidaris crassispina*) inhabits intertidal and subtidal shallow waters along the northern coast of the South China Sea [8]. Its gonads are rich in bioactive compounds and are highly nutritious [9,10,11,12,13], making it widely appreciated [14]. The Daya Bay area in Guangdong Province is rich in purple sea urchin resources and is an important production area in China. However, owing to overfishing, habitat destruction, and climate change, the population of purple sea urchins has declined greatly. Researchers studying purple sea urchins have focused on their resource distribution, population structure, and nutritional value [15,16]. Studies on the feeding habits of purple sea urchins are relatively rare, with only a few studies by researchers, such as Qin and Mo, investigating the impact of purple sea urchins on the food web of large benthic animals in Daya Bay [8,17]. Understanding the feeding habits of purple sea urchins is important for enhancing their populations and restoring their habitats.

Understanding the diet and behavior of animals is crucial for comprehending their behaviors and survival strategies [18]. Historically, researchers have used stomach content analysis (SCA) to elucidate the trophic interactions of predatory fish, but this method can reveal only the diet of an animal over a short period and does not reflect the animal’s overall diet [19]. Stable isotopes are globally recognized tools for reconstructing diets, characterizing nutritional relationships, elucidating resource allocation patterns, constructing food webs [20,21], and providing insights into the assimilation of prey over longer periods. Carbon (C) and nitrogen (N) stable isotopes are the most commonly used elements in marine food web studies [22]. The ratio of ^15^N to ^14^N (expressed as δ^15^N) reveals the progressive enrichment of nutrients and can be used to assess the trophic position of consumers. The ratio of carbon isotopes varies greatly among primary producers with different photosynthetic pathways (e.g., C3 vs. C4) but changes little during nutrient transfer, making δ^13^C useful for determining the original source of dietary carbon [23]. While stable isotope techniques offer great advantages in animal dietary analysis, the mathematical uncertainties associated with isotope mixing model analyses [24] can lead to the potential misuse or misinterpretation of the results [25]. Moreover, 16S high-throughput sequencing has become an important tool for analyzing the composition and variability of microbial communities in complex environments and can be used to identify and characterize the structure of microbial communities [26]. Similarly, 18S high-throughput sequencing targets eukaryotes in the animal gut, providing insights into the construction of phylogenetic trees and the composition of the organisms consumed by animals. This technique has been successfully applied to reveal the composition and diversity of microbial communities in the guts of shrimp [27] and mandarin fish [28].

Furthermore, high-throughput sequencing technology has attracted increasing attention because of its powerful potential in animal dietary analysis [29]. Compared with traditional dietary analysis methods, analyzing residual feces and materials in the gut via 16S/18S DNA sequencing can yield more accurate dietary analysis results [30]. For example, Pan and Qin used metagenomics to determine the diet of yellowfin sea bream, revealing its dietary range [31]; Chai and others used metagenomics to reveal the dietary structure of ruminants such as goats [32]; and Junhyung Kim and others used metagenomics to reveal the dietary range of domestic poultry such as chickens [33]. Therefore, we can explore animal diets by combining stable isotope and high-throughput sequencing technologies.

Preliminary investigations revealed that purple sea urchins are distributed in both the stone and algal areas of Daya Bay, with a greater density in the stone area than in the large algal area. This raises important scientific questions: What do the purple sea urchins in the stone area feed on? What is the dietary range of purple sea urchins? Therefore, we focused on purple sea urchins and employed stable isotope technology and 16S rDNA and 18S rDNA high-throughput sequencing techniques to conduct a systematic study on the feeding habits and gut microbiota community structure of purple sea urchins in the stone and algal areas of the central islet sea area of Daya Bay. We aimed to elucidate the feeding habits and dietary range of purple sea urchins. The results of this study provide a theoretical basis for the restoration of wild purple sea urchin resources and the selection of areas for restocking and release.

## 2. Materials and Methods

### 2.1. Sample Collection and Processing

The sea urchins used for the experiment were collected on 29 October 2021, from the Dalajia Stone area of Daya Bay (Figure 1). The sampling area was divided into two zones on the basis of differences in the habitat of *Strongylocentrotus purpuratus*: the algae zone (area A, with abundant algae but fewer sea urchins) and the stone zone (area S, with fewer algae and mostly exposed rocks), as depicted in Figure 2. The potential food sources for *S. purpuratus* generally include plankton, macroalgae, sediment, and attached diatoms [8]. Owing to the availability of samples from different sampling areas, plankton, *Padina* sp./spp. (a type of brown algae), and sediment were collected from the stone zone as potential food sources. In the algae zone, sediment, *Sargassum hemiphyllum* (a type of brown algae), plankton, and diatoms attached to macroalgae were collected as potential food sources. Three sampling sites were established in each zone, and *S. purpuratus* samples were obtained by divers. A total of 36 samples were collected, and large algae and surface sediment were collected within 5 m of the three sampling sites and their adjacent areas. Plankton were collected using a Type II plankton net and a Type III phytoplankton net towed slowly behind the boat.

In situ measurements of salinity (Sal), temperature (T), *pH*, and dissolved oxygen (DO) were conducted at each sampling site using an American Sea-Bird Multi-Parameter Water Quality Meter (YSI-556MPS), with three replicate measurements taken and the average value recorded. Owing to the shallow water depth at the sampling points, only surface seawater was collected, with 1000 mL water samples taken at each site. The physicochemical parameters of water quality mostly included measurements of ammonia nitrogen (N-NH_4_^+^, bromine oxidation method), nitrite (N-NO^2−^, naphthylethylenediamine spectrophotometric method), nitrate (N-NO^3−^, zinc-cadmium reduction method), and reactive phosphate (P-PO_4_^3−^, phosphomolybdenum blue spectrophotometric method), all of which were measured in accordance with the “Marine Monitoring Specifications—Seawater Analysis” (GB 17378.4-2007). The total nitrogen (TN) and total phosphorus (TP) contents were determined via alkaline potassium persulfate digestion.

### 2.2. Sample Preparation

The shell diameter and shell height of the collected *S. purpuratus* samples were measured using a Vernier caliper (accurate to 0.1 mm), and the samples were weighed using an electronic balance (accurate to 0.1 g). The gonads and Aristotle’s lantern muscles were dissected from the sea urchins. Large algae were washed 2–3 times under sterile conditions with a 0.45 μm GF/F membrane (preburned at 450 °C for 6 h to remove inorganic carbon) to obtain attached diatoms. The sediment was rinsed 3–5 times with sterile seawater, allowed to settle, and then filtered through a 500-mesh sieve. Plankton were allowed to settle for 24 h and were filtered through an 80-mesh sieve using tweezers to remove impurities. Water samples were filtered through a 0.2 μm GF/F membrane (preburned at 450 °C for 6 h to remove inorganic carbon). The sea urchin gonad samples, large algae, and plankton were wrapped in aluminum foil and rapidly frozen at −80 °C until they were stable in shape. All of the samples were then freeze-dried for 48 h in a −55 °C freeze dryer, transferred to 1 mL centrifuge tubes, ground into a fine powder using a grinding machine, treated with 1 mol/L hydrochloric acid solution to remove carbon, dried, and stored for isotope analysis.

### 2.3. Stable Isotope Analysis

The stable isotope ratios of carbon (δ^13^C) and nitrogen (δ^15^N) in the samples were determined using a stable isotope ratio mass spectrometer (Isoprime 100) and an elemental analyzer (PYRO Cube). The precision for both the δ^13^C and δ^15^N analyses was set at 2‰ (parts per thousand). The obtained stable isotope ratios were calculated using the following Formula (1):(1)$$\delta X=\left(\frac{R_{sample}}{R_{standard}}-1\right)\cdot 10^{3}$$

In the equation, *X* represents either ^13^C or ^15^N, and *R* denotes the ratio of ^13^C/^12^C or ^15^N/^14^N. The standard for carbon isotopes is *Belemnitella americana*, and the standard for nitrogen isotopes is atmospheric nitrogen. A standard sample test was required for every ten samples analyzed to ensure the accuracy of the experimental results and the stability of the instrument.

### 2.4. High-Throughput Sequencing of 18S rDNA

The nitrocellulose filter was cut into pieces with sterile scissors. Total DNA was extracted from the nitrocellulose membrane using a Power Soil© DNA Isolation Kit (MOBIO, Carlsbad, CA, USA) following the manufacturer’s instructions. The purity of each DNA sample was quantified and assessed using a NanoDrop ND-1000 spectrophotometer (NanoDrop Technologies, Wilmington, DC, USA), and the quality of the extracted DNA was checked by electrophoresis on a 1% agarose gel. All of the DNA samples were stored at −80 °C until sequencing.

The V4 variable region of the 18S rDNA gene was amplified using the universal primers TAReuk454FWD1F (5′-CCA GCA SCY GCG GTA ATT CC-3′) and TAReukREV3R (5′-ACT TTC GTT CTT GAT YRA-3′). PCR amplification was performed using the Phusion^®^ High-Fidelity PCR Master Mix (New England Biolabs). The V4 region of the 18S rDNA gene was subjected to three independent replicate PCRs with the following reaction conditions: initial denaturation at 98 °C for 1 min, annealing at 50 °C for 30 s, denaturation at 98 °C for 10 s, 30 cycles of extension at 72 °C for 60 s, and a final extension at 72 °C for 5 min. The PCR products from the same sample were mixed and separated by electrophoresis on a 1% agarose gel. The PCR products were purified using the TIANgel Purification Kit (TIANGEN Biotech, Beijing, China) according to the manufacturer’s instructions. The PCR products were quantified and assessed for quality using the QuantiFluor™-ST blue fluorescence quantification system (Promega Corporation, Madison, WI, USA). Finally, an Illumina PE250 library was constructed for sequencing.

The paired-end (PE) reads obtained from Illumina PE250 sequencing were assembled using FLASH software (Version 1.2.11., John Hopkins University, Baltimore, MD, USA). High-throughput sequencing data were preprocessed using QIIME v1.9.1, and sequences with low quality scores and short lengths were discarded. The chimeric sequences were subsequently detected and filtered out via the ‘parallel_identify_chimeric_seqs.py’ and ‘filter_fasta.py’ commands. Nonredundant sequences were clustered at 97% similarity using the Uclust algorithm, partitioning them into operational taxonomic units (ASVs). The most abundant sequence in each ASV was selected as the representative sequence for that ASV. The representative sequences of each eukaryotic ASV were matched with the corresponding microbial ribosomal database (bacteria and protists—SILVA database; fungi—UNITE database) to select sequences of micro-organisms for further analysis. The eukaryotic ASV table used in this study was generated by removing fungal and animal sequences. The R package vegan was used to perform rarefaction analysis for the filtered ASVs to determine whether the obtained sequences covered most of the complete gut microbiome (rarefaction curves). The rarefaction curves gradually flattened, confirming that the sequencing depth of the sampling dataset was sufficient. These standardized ASV abundance tables were used for all subsequent analyses.

### 2.5. 16S rDNA High-Throughput Sequencing

The amplification of the V3–V4 variable region of the 16S rDNA gene was performed using the universal primers 341F (5′-CCTACGGGNGGCWGCAG-3′) and 806R (5′-GGA CTACHVGGGTATCTAAT-3′). PCR amplification was carried out using the Phusion^®^ High-Fidelity PCR Master Mix (New England Biolabs, Ipswich, MA, USA). The PCRs for the V3–V4 region of the 16S rDNA gene involved three independent replicate PCRs with the following conditions: initial denaturation at 94 °C for 2 min, annealing at 50 °C for 30 s, 30 cycles of denaturation at 98 °C for 10 s, extension at 68 °C for 60 s, and a final extension at 68 °C for 5 min. The PCR products from the same sample were mixed and separated by electrophoresis on a 1% agarose gel. The PCR products were purified using a TIANgel Purification Kit (TIANGEN Biotech, Beijing, China) according to the manufacturer’s instructions. The PCR products were quantified and assessed for quality using the QuantiFluor™-ST blue fluorescence quantification system (Promega Corporation, Madison, WI, USA). Finally, an Illumina PE250 library was constructed for sequencing.

### 2.6. Statistical Analysis

The data obtained were analyzed using SPSS Statistics 25.0. Grouped data t-tests were employed to compare the differences in δ^13^C or δ^15^N between stone and algal zone sea urchins and their potential food sources, as well as the differences in δ^13^C or δ^15^N in the gut contents of sea urchins from the stone and algal zones. The C/N ratios of food sources for sea urchins from both areas were compared, with a significance level of *p* < 0.05. The R language SIMMR package was used to analyze the contributions of potential food sources from different habitats to the diets of sea urchins.

The CCS sequences were derived from the original data, and barcode identification and length filtering were performed for the CCS sequences. The dada2 method in QIIME2 2020.6 software was used for denoising and removing chimeric sequences to obtain the final effective data (nonchimeric reads). Taxonomic analysis was then performed for the samples at various classification levels to construct community structure diagrams at the phylum, class, order, family, genus, and species levels. The alpha diversity indices of the micro-organisms, including the Chao1 index, ACE index, Shannon–Wiener index, and Simpson index, were subsequently calculated via the vegan package in R. The VennDiagram package in R was used to visualize the Venn diagrams, indicating the relationships between the microbial ASVs of different attached biota. The beta diversity of the microbial communities from the attached biota areas was analyzed using principal coordinate analysis (PCoA).

## 3. Results and Analysis

### 3.1. δ^13^C and δ^15^N Values and C/N Ratios of Potential Food Sources

The average shell diameter of sea urchins in the stone area was 39 ± 4.40 mm, with an average weight of 27.84 ± 5.06 g. The range of δ^13^C values for sea urchins was −11.42‰ to −8.17‰, whereas the mean δ^15^N values ranged from 9.15‰ to 10.31‰. In contrast, the sea urchins in the algal area had an average shell diameter of 48.54 ± 11.06 mm and an average weight of 60.91 ± 30.53 g. The δ^13^C values of these sea urchins ranged from −13.97‰ to −12.44‰, and the mean δ^15^N values ranged from 8.75‰ to 10.14‰ (Figure 3, Table 1). The one-way ANOVA results for stable isotopes in sea urchins from both areas indicated a significant difference in δ^13^C (*p <* 0.05) between the two areas but no significant difference in δ^15^N (*p >* 0.05).

The δ^13^C and δ^15^N values of the gut contents of *S. purpuratus* from Daya Bay ranged from −20.1‰ to −10.32‰ and from 7.64‰ to 8.99‰, respectively, with the highest values found in plankton and the lowest in *Padina* and *Sargassum* (Figure 4). The results of the grouped data t-tests indicated that there was no significant difference in the δ^13^C values of plankton between the stone and algal zones (*p >* 0.05), but significant differences were detected in the sediments, *Padina*, *Sargassum*, and attached diatoms (*p <* 0.05). The Δδ^13^C value between the attached diatoms in the stone and algal zones reached 17.66‰. There were significantly more plankton and sediments collected from the algal zone than from the stone zone. There were significantly more attached diatoms and *Padina* collected from the stone zone than from the algal zone.

In the Daya Bay area, in addition to plankton, the sediment of stone areas, attached diatoms, and *Sargassum* were greater than those in the algal areas (Figure 5). On average, the C/N ratio of food sources in the stone areas was greater than that in the algal areas. However, there were significant differences (*p* < 0.05) between the stone and algal areas in terms of plankton, attached diatoms, and *Sargassum*, whereas the differences in sediment between the two areas were not significant (*p* > 0.05).

### 3.2. Stable-Isotope-Ratio-Based Analysis of the Diet of the Purple Sea Urchin

The contributions of potential food sources from different habitats to the diet of sea urchins were analyzed via the SIMMR package in R (Figure 6). The results indicated that, in both stone and algal areas, the contribution of sediment to sea urchins exceeded 44%. Compared with that in the stone area, the contribution of algae to sea urchins in the algal area was 40%, whereas the contribution of algae to sea urchins in the stone area was only 32.1%, which was lower than that in the stone area.

### 3.3. Analysis of the Diet Composition and Relative Abundance of Purple Sea Urchins Based on 18S

Analysis of the shared eukaryotic organisms in the gut contents of sea urchins from the stone and algal areas revealed (Figure 7) that, in the stone area, the gut contents of sea urchins contained 36 amplifier sequence variants (ASVs), with 6 ASVs being shared between the two areas. In the algal area, the gut contents of the sea urchins contained 11 ASVs.

Through the classification and statistical analysis of eukaryotic organisms at the phylum and genus levels, it was found that, in the algal area, the dominant eukaryotic groups in the gut contents of sea urchins were the phyla *Molluska* and *Ascomycota*, with relative abundances greater than 86%. In contrast, the dominant eukaryotic group in the gut contents of sea urchins from the stone area was the phylum *Arthropoda*, which accounted for more than 76% of the microbial community in the gut. At the genus level, the dominant eukaryotic groups in the gut contents of sea urchins from the algal area were the genera *Clypeomorus* and *Candida*, with relative abundances greater than 86%. In the stone area, the dominant eukaryotic group in the sea urchin gut contents was the genus *Temora*, accounting for more than 74% of the microbial community (Figure 8).

According to the classification and statistical analysis of eukaryotic organisms in the environment at the phylum level, in the algal area, the dominant eukaryotic groups in the sediment were the phyla *Molluska* and *Arthropoda*, with relative abundances greater than 72%. In the water environment, the dominant eukaryotic groups were *unclassified Eukaryota* and *Arthropoda*, with relative abundances greater than 39%. In the stone area, the dominant eukaryotic groups in the sediment were the phyla *Arthropoda*, *Molluska*, and *Cnidaria*, with relative abundances greater than 82%. In the water environment, the dominant eukaryotic group was the phylum *Basidiomycota*, with a relative abundance of 88.6% (Figure 9).

The dilution curve indicates that the eukaryotic sequences obtained from the sea urchin gut via 18S rDNA reached a plateau (Figure 10), suggesting that the sequencing results covered the species diversity of the samples, enabling data analysis. Alpha diversity analysis of the eukaryotic organisms in the guts of sea urchins from different habitats (Figure 11, Table 2) revealed significant differences in the ACE and Chao1 indices (*p* < 0.05), whereas the Shannon and Simpson indices were not significantly different (*p* > 0.05). This finding indicated that there was a significant difference in the richness of the feeding organisms between the two regions of sea urchins, but there was no significant difference in biodiversity. Principal coordinate analysis (PCoA) of the eukaryotic organisms in the guts of sea urchins from the two regions (Figure 12) revealed that pc1 contributed 43.42% of the sample differences, and pc2 contributed 38.44%. The confidence ellipses of the eukaryotic organisms overlapped to some extent, suggesting that the differences in the composition of eukaryotic organisms in the gut contents of sea urchins between the two regions were not significant.

### 3.4. Diversity and Differential Analysis of the Sea Urchin Gut Microbiome

An analysis of the commonalities of eukaryotes in the gut contents of sea urchins from stone and algal areas (Figure 13) revealed 425 amplicon sequence variants (ASVs) in the gut contents of sea urchins from the stone area, with 158 ASVs being shared between the two areas. In contrast, there were 197 ASVs in the gut contents of sea urchins from the algal area.

Through classification statistics at the phylum and genus levels, it was found that, at the phylum level, the dominant groups of the gut microbiota in sea urchins from both the algal and stone areas were *Bacteroidota* and *Proteobacteria*, with relative abundances exceeding 50%. At the genus level, the dominant microbial populations in the guts of sea urchins from both areas were *Roseimarinus* and *Photobacterium*, with relative abundances exceeding 43% (Figure 14).

Additionally, classification statistics at the phylum level for environmental micro-organisms revealed that, in both the algal and stone areas, the dominant microbial groups in the sediment and water environments were *Bacteroidota* and *Proteobacteria*, with relative abundances exceeding 62% (Figure 15).

The dilution curve indicates that the 16S rDNA sequences of the gut microbiota and eukaryotic organisms obtained from sea urchins reached a plateau (Figure 16), suggesting that the sequencing results adequately covered the species diversity of the samples and were suitable for data analysis. Alpha diversity analysis of the gut microbiota in sea urchins from different habitats (Figure 17, Table 3) revealed significant differences in the ACE and Chao1 indices (*p* < 0.05), whereas the Shannon and Simpson indices showed no significant differences (*p* >0.05), indicating that there was a significant difference in the richness of the feeding organisms between the two regions of sea urchins but no significant difference in biodiversity. PCoA of the gut microbiota in sea urchins from the two regions (Figure 18) revealed that pc1 contributed 34.12% of the sample differences, and pc2 contributed 19.19%, indicating that there were significant differences in the composition and structure of the gut microbiota between the sea urchins from the different regions.

## 4. Discussion

### 4.1. Combining Stable Isotope Analysis with High-Throughput Sequencing Technology Can Provide a More Comprehensive Understanding of an Animal’s Feeding Habits

Stable isotope analysis has been established as a valuable tool for reconstructing diets and constructing food webs because it involves the use of stable isotopes such as ^13^C and ^15^N to identify the gut microbiota and its ecological and physiological characteristics [34]. Therefore, we utilized ^13^C and ^15^N to reveal the potential food sources of purple sea urchins in both stone and algal areas, such as plankton and sediment. Additionally, research has indicated that stable isotope data are suitable for demonstrating general differences in trophic ecology, but they cannot provide detailed information for the exact prey items consumed by organisms [35]. Furthermore, our isotope analysis results did not reveal certain taxa, such as *Molluska* and *Arthropoda*, which were identified in the high-throughput sequencing results. This discrepancy is likely because short-term dietary fluctuations in purple sea urchins may not be reflected in stable isotopic signatures. The isotopic results represented an average purple sea urchin diet over a period of approximately three months rather than an instantaneous snapshot [36].

Over the past few decades, critical breakthroughs in nucleic acid sequencing technology and molecular techniques have propelled 16S rDNA gene sequencing into a cornerstone of modern microbial ecology [37]. As a result, 16S rDNA gene microbial community analysis has become a mainstream practice among experts and scholars, serving as the foundation for several molecular techniques. Additionally, the amplification of the 16S rDNA gene has been widely used to study the composition and dynamics of microbial communities in the gut [38]. As demonstrated by Kerry Jo Lee and colleagues, the analysis of wild primate gut microbiomes can be used to determine diet [39]. Furthermore, Brian D. and colleagues reported that the gut microbiota of mammals can influence their diet [40]; however, this influence is not only related to the composition of the gut microbiota but is also a critical factor in the coevolution of animals and their gut microbiota. The gut microbiota is critical in animals adapting to different diets because it provides essential metabolic pathways for the host [41].

To our knowledge, 16S rDNA sequencing is the most commonly used method for estimating differences between bacterial species. Most studies on 16S rDNA are based on a single gene, specifically the small subunit ribosomal RNA (rRNA) gene, which is encoded by highly conserved 16S ribosomal DNA (rDNA) [42]. Furthermore, this gene is characterized by high expression, stability, and conservation, making it suitable for the universal detection of bacteria [43]. Therefore, we can utilize the high-throughput sequencing of 16S rDNA to comprehensively describe the bacterial diversity and community composition, and, by integrating the dietary habits of animals, we can investigate the impact of food on the gut microbiota of animals. This approach enables us to gain a deeper understanding of their dietary preferences.

Additionally, owing to the high specificity and sequence conservation of the 18S rDNA gene, it has become the most commonly used marker for exploring the community structure of eukaryotic organisms in both aquatic and terrestrial environments [44]. To some extent, the coupling of stable isotope analysis with the metagenomic sequencing of microbial communities enables us to identify the diet and lifestyle of biological organisms. The combined use of these methods may become the most effective tool for analyzing biological diets in the future. Researchers have reported that 18S rDNA is a widely used ABS marker in eukaryotic organisms because of its highly conserved primer-binding sites and sufficient variability within species, which reflects phylogenetic relationships [45]. Additionally, the 18S rDNA gene has been successfully applied to a variety of sample types, including human, host, and insect vectors [46]. Furthermore, 18S rDNA is one of the most conserved DNA sequences and is characterized by a moderate length and redundant biological information. It has a relatively low nucleotide substitution rate and evolutionary rate. Moreover, the conserved regions of 18S rDNA are suitable for reconstructing phylogenetic trees for all organisms, whereas the variable regions are used for distinguishing genera or species [47]. To date, T. Makwanise and colleagues have utilized 18S rDNA to analyze the gut contents of poultry to identify their diet (*Gallus gallus bastricicus*) [48]. Therefore, 18S rDNA is considered a reliable and informative marker for species identification and genotyping as it can be used to detect eukaryotic organisms in the animal gut, enabling us to understand the types of organisms that animals consume and determine their diet [46].

### 4.2. Habitat Environmental Characteristics Drive Changes in the Gut Food Composition of Purple Sea Urchins

Different regions possess distinct food sources [49,50,51,52], which also determine the variety of food items that animals consume. We found that the δ^13^C range of potential food sources for purple sea urchins in the stone and algal areas of Daya Bay ranged from −11.42‰ to −8.17‰ (*p* < 0.05), indicating a significant difference. Additionally, we observed that purple sea urchins fed on *Gracilaria* in the stone area and *Sargassum* in the algal area. This suggested that, similar to *Tripneustes* and *Eucidaris*, purple sea urchins consume a variety of prey on the basis of the availability of food sources in the environment [53]. Furthermore, the δ^13^C values of potential food sources such as sediments, *Gracilaria*, *Sargassum*, and attached diatoms in the stone and algal areas are significantly different. The reason for this could be that small herbivores and detritivores remove filamentous epiphytes and sediments from the surfaces of large algae, thereby promoting the growth of fleshy algae [54]. This results in a higher sediment content in the algal area than in the stone area, an absence of attached diatoms in the stone area, and a low abundance of attached diatoms in the algal area. This is partly due to the competitive disadvantage of attached diatoms in relation to other algal species [55]. On the other hand, competition among large algae can crucially impact the environment [56,57,58], leading to large differences in the δ^13^C values of potential food sources for purple sea urchins in the two areas. Additionally, the C/N ratio is an important indicator of the nutritional value of organic matter; a lower C/N value indicates a higher nitrogen content and, consequently, a greater relative nutritional value [59]. As shown in Figure 5, the C/N ratio in the algal area was significantly lower than that in the stone area. These findings indicate that the nutritional value of the food consumed by the purple sea urchins in the algal area was greater than that of the food consumed by the purple sea urchins in the stone area.

From the perspective of potential food sources contributing to the diet of sea urchins, the main food source for sea urchins in both areas are sediments. In the stone area where large algae are scarce, sediments contribute up to 55.2% of the sea urchin’s diet, whereas, in the algal area where large algae are abundant, sediments contribute 44.3% of the sea urchin’s diet, which is greater than that of *Sargassum* (19.2%). These findings indicate that food abundance in the environment is not the only factor influencing sea urchin feeding habits. For example, the feeding preferences and consumption rates of sea urchins are often influenced by the nutritional content of their food sources [60]. In the analysis of the sediments (as shown in Figure 11), the major eukaryotic organisms identified were from the phyla *Molluska* and *Arthropoda*. This finding aligns with findings from other studies on *Arbacia lixula*, which have shown greater total proteolytic activity and acidic proteolytic activity [61]. These findings suggest that the feeding habits of this species are not closely related to herbivory, leading to the hypothesis that the diet of purple sea urchins is likely omnivorous. Additionally, in *Sargassum* beds, the main component of the gut contents of *T. depressus* is brown macroalgae (*Sargassum* and *Dictyota*), followed by red and green macroalgae [62]. This contrasts with the primary diet of sediment observed for the purple sea urchins from both regions, which may be because sea urchins exhibit food preferences when presented with a variety of food sources [53].

### 4.3. Diet and Environment Can Influence the Composition and Diversity of the Gut Microbiota in Sea Urchins by Inducing Changes in Their Feeding Preferences

Our results indicated that purple sea urchins have a unique preference for animal food sources. The composition of eukaryotic organisms in the gut of purple sea urchins is closely related to the algal area and the stone area. In the algal area, the eukaryotic organisms in the gut of the purple sea urchin were mainly from the genus *Trochus* (57.37%), whereas, in the stone area, the eukaryotic organisms were mainly from the genus *Temora* (74.62%). This difference may be because the purple sea urchin is an omnivorous animal that consumes a wide range of potential food sources [63]. Additionally, researchers have reported that the genera *Trochus* and *Temora* are distributed in sediments [64,65]. Combining the relative contributions of potential food sources in the algal area, as shown in Figure 6, it is likely that the purple sea urchin prefers mollusks and arthropods when selecting food items. Furthermore, a comparison of the gut contents of sea urchins from different areas revealed that the biodiversity of both micro-organisms and eukaryotic organisms was much greater in the stone area than in the algal area. This could be related to the fact that epiphytic mosses can damage algae, leading to a reduction in biomass in the algal area [66]. Additionally, there is evidence from other scholars that sea urchins can have a biocidal effect on algae [67,68]. The reduction in biomass in the algal area, possibly due to the biocidal effect of sea urchins on algae, could lead to an increase in the biodiversity of organisms in the gut of sea urchins in the stone area. This is because the loss of algae in the algal area may force sea urchins to seek alternative food sources, which could include a wider range of organisms found in the stone area. The greater biomass of gastropods in the algal area could be attributed to their preference for parts of brown algae, which they can adapt to because of the substrate characteristics of the algae (such as shape or complexity). This adaptation enables gastropods to thrive in specific environmental conditions and form morphologically functional groups that are well-suited to their habitat [69]. Furthermore, the number of attached diatoms was significantly greater in stone areas than in algal areas (as shown in Figure 4), and copepods have exhibited significant behavioral adaptations to different diatom species [70]. These results indicated that the copepod biomass is greater in stone areas than in algal areas. These findings collectively influence the differences in feeding among purple sea urchins in the two regions. In summary, purple sea urchins in Daya Bay exhibit differences in feeding between stone and algal areas, with a preference for omnivory, which is related to the feeding preferences of sea urchins and the presence of different species and their biomasses in different regions.

Our research results suggest that the structure and function of the gut bacterial community in purple sea urchins are strongly influenced by diet. There are large differences in the gut microbiota of aquatic organisms that feed on different foods. Studies have shown that certain algal species produce secondary metabolites (such as xanthophylls and phenolics) to prevent herbivorous organisms from feeding on them [71], which could lead to a preference for animal-based food sources in sea urchins from both areas, which, in turn, may affect the number of gut microbes. Additionally, some researchers have documented a negative correlation between phenolic concentrations and the feeding preferences of herbivores [72]. However, whether these factors affect sea urchin herbivory remains to be investigated. The feeding preferences of herbivores can also be influenced by the nutritional quality of the algae [73]; however, the extent to which these factors influence sea urchin herbivory requires further substantiation.

On the basis of the feeding preferences of purple sea urchins, we can infer that the preference for animal-based food sources may be due to the intrinsic factors of the algal species. This preference leads to differences in the structure and composition of the gut microbiota. We found that sea urchins in the stone area have a high diversity of gut organisms at all taxonomic levels, which may be related to the number of sea urchins in the area. The number of sea urchins can affect the dominant algal groups in a region, which, in turn, influences the species and abundance of organisms in that area [74]. Concurrently, research findings indicate that sea urchins greatly alter shallow-water habitats by transforming upright brown algae beds into bottom areas devoid of upright algae around temperate and subtropical stones [75,76]. These findings suggest that the gut microbial community of sea urchins is influenced by their dietary sources.

In aquatic animals, the composition, activity, and interactions of the gut microbial community can influence the amount of energy extracted from the diet and are crucial in the metabolism of dietary substrates and the regulation of the immune system [77]. PCoA of the gut microbiota and eukaryotic organisms in sea urchins from stone and algae bed areas in Daya Bay revealed distinct differences in microbial composition between the two regions. The phyla *Bacteroidetes* and *Proteobacteria* were found to be prevalent, and these two phyla were identified as major components in the study of the gut microbiota structure, which aligns with findings from similar research on the gut microbiota of crustaceans [78]. Moreover, *Bacteroidetes* are primarily studied as beneficial bacteria [79]. For example, *Bacteroidetes* can release biomolecules such as polysaccharide A and sphingolipids, which are critical for regulating the host immune system in both health and disease [80]; *Proteobacteria*, on the other hand, have opposite effects. The *Proteobacteria* phylum is one of the largest in the gut microbiota and includes numerous opportunistic pathogens [81]. On the basis of the experimental results, in the stone area, the *Proteobacteria* phylum has become the dominant group in the sea urchin gut, leading to a greater incidence of disease than in sea urchins in the algal area. Furthermore, a taxonomic classification at the phylum level of the environmental micro-organisms associated with sea urchins in both the algal and stone areas revealed that the dominant groups in the sediments of both regions are primarily the *Bacteroidetes*, *Proteobacteria*, and *Verrucomicrobia* phyla, which are roughly consistent with the dominant groups in the gut microbiota of the sea urchins in these two regions. This finding aligns with observations that sediment is a potential primary food source for sea urchins (Figure 6).

However, the dominant groups in the water environment of both regions are the *Proteobacteria*, *Cyanobacteria*, and *Bacteroidetes* phyla, which differ from the microbial composition in the sea urchin gut. *Cyanobacteria* are abundant in the water environment but scarce in the sea urchin gut, suggesting that, similar to other sea urchins, they may alter their feeding behavior according to environmental conditions, acting as passive filter feeders that capture drifting algae [82]. At the genus level, the dominant genera in the sea urchin gut from both regions were the genera *Roseobacter* and *Photobacterium*. The *Roseobacter* genus is a type of bacterium within the *Bacteroidetes* phylum [83]. Research has also indicated that the *Roseobacter* genus is associated with the assessment of gonad freshness in sea urchins. The freshness of the gonads is inversely proportional to the content of *Photobacterium* and directly proportional to the content of *Roseobacter* [84]. *Photobacterium* is often involved in the formation of putrescine and cadaverine in fish and shrimp [85]. These findings suggest that sea urchins in stone areas have a poorer health status than those in algal areas, and that the gut microbiota is influenced by the habitat environment.

## 5. Conclusions

We employed stable isotope techniques and high-throughput sequencing to analyze the diets of purple sea urchins in the stone and algal areas of Daya Bay to obtain high-resolution species information on eukaryotic and microbial communities. The stable isotope results revealed significant differences in the δ^13^C values in the guts of sea urchins from the two areas (*p* < 0.05), whereas the δ^15^N values were not significantly different (*p >* 0.05). The primary food source for sea urchins in both areas is sediments. The sequencing of the 18S rDNA gene revealed that, in the guts of purple sea urchins from the algal area, most of the detected sequences belong to the phylum *Molluska*. In contrast, in the guts of sea urchins from the stone area, most belong to the phylum *Arthropoda*. The sequencing of the 16S rDNA gene revealed that, in the guts of purple sea urchins from the algal area, *Bacteroidetes* were the dominant group, whereas, in the guts of sea urchins from the stone area, *Proteobacteria* were the dominant group. Diversity analysis revealed significant differences in the numbers of microbial and eukaryotic organisms in the guts of sea urchins from the two areas (*p* < 0.05), with the stone area exhibiting greater microbial and eukaryotic diversity than the algal area.

However, this study was limited to the analysis of gut contents from purple sea urchins collected in autumn and did not include samples from a full year or consider the differences in dietary preferences due to age differences in sea urchins. Future research should collect gut contents from sea urchins at different developmental stages and collect samples at various times throughout the year to perform comparative analyses. This will increase the accuracy of the experimental results by enabling a more comprehensive understanding of the seasonal and age-related variations in the diets of purple sea urchins. The study of sea urchin feeding habits is highly valuable for marine ecology, resource management, biodiversity conservation, and the sustainable development of the marine economy. This study contributes to the development of effective conservation measures and management policies that promote the sustainable use of marine resources and the health of ecosystems. In the future, this research is expected to make significant contributions to the sustainable development of human society and the conservation of marine resources.

## Figures and Tables

**Figure 1 biology-13-00623-f001:**
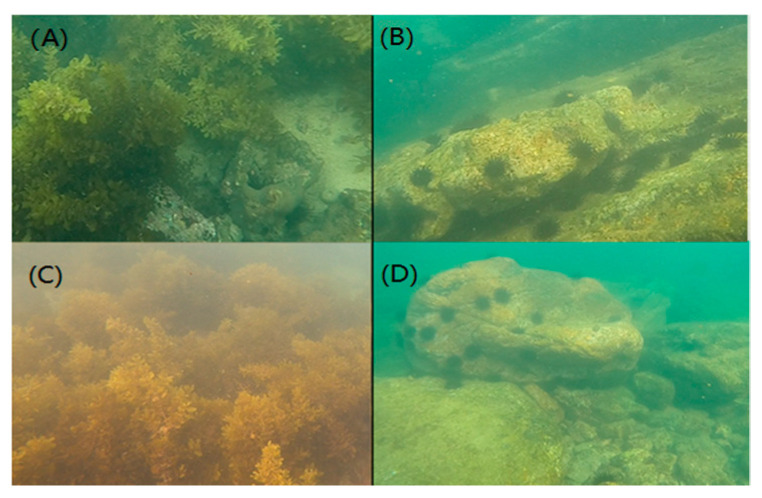
Overview of the sampling areas. (**A**,**C**) represent the algal zone. (**B**,**D**) represent the stone zone.

**Figure 2 biology-13-00623-f002:**
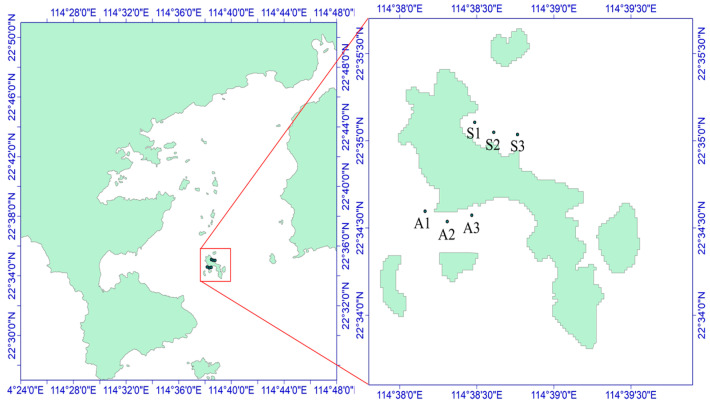
Schematic diagram of the sampling sites in the Daya Bay area. A1–A3 and S1–S3 are six sampling sites, of which S1–S3 is the northern part of Daya Bay and A1–A3 is the western part of Daya Bay.

**Figure 3 biology-13-00623-f003:**
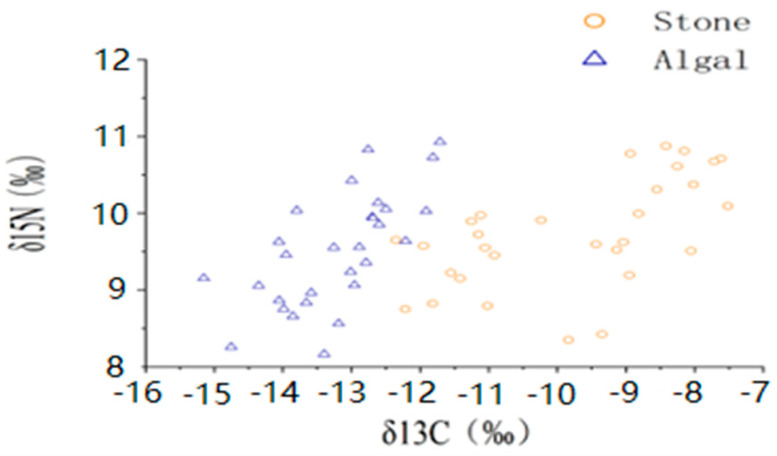
Isotopic characteristics of purple sea urchins from different regions.

**Figure 4 biology-13-00623-f004:**
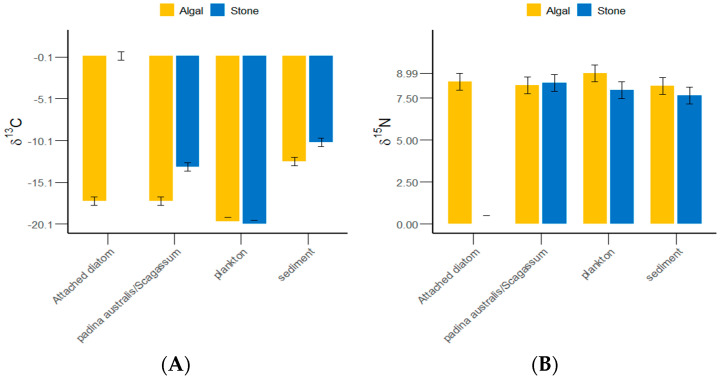
δ^13^C and δ^15^N values of potential food sources for sea urchins in the stone and algal zones of Daya Bay. (**A**) Comparative analysis of δ^13^C in sediments, Padina, Sargassum, and attached diatoms. (**B**) Comparative analysis of δ^15^N in sediments, Padina, Sargassum, and attached diatoms.

**Figure 5 biology-13-00623-f005:**
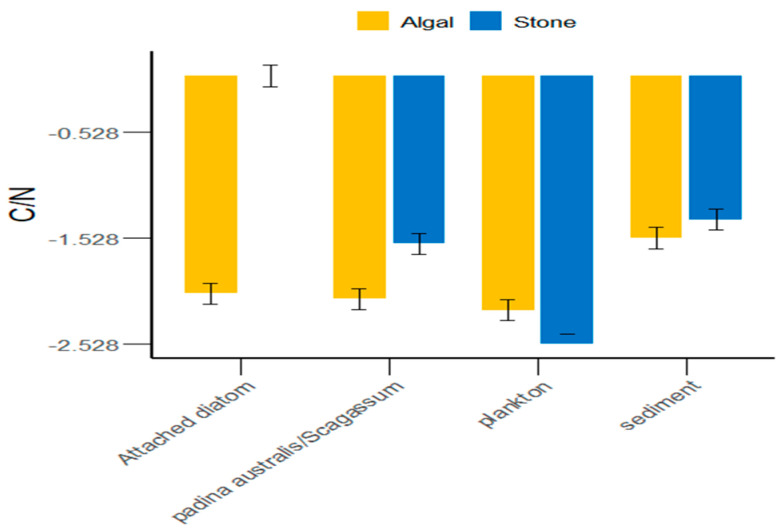
C/N ratios of potential food sources for sea urchins in the stone and algal areas of Daya Bay.

**Figure 6 biology-13-00623-f006:**
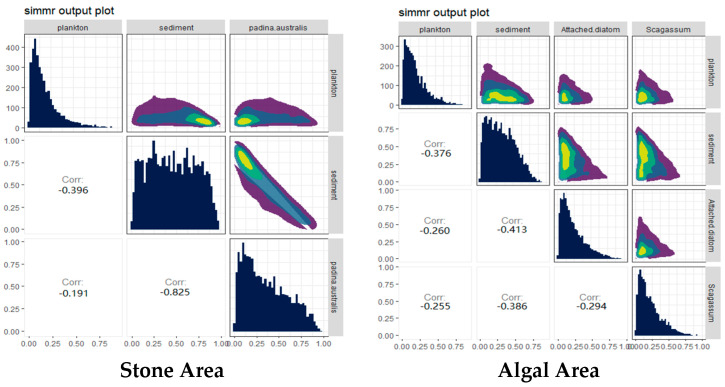
Relative contributions of potential food sources to the purple sea urchin in the two areas. The colors in the image mainly represent the data density. The color ranges from purple to dark blue to light green and yellow, indicating that the distribution density of data points ranges from low to high.

**Figure 7 biology-13-00623-f007:**
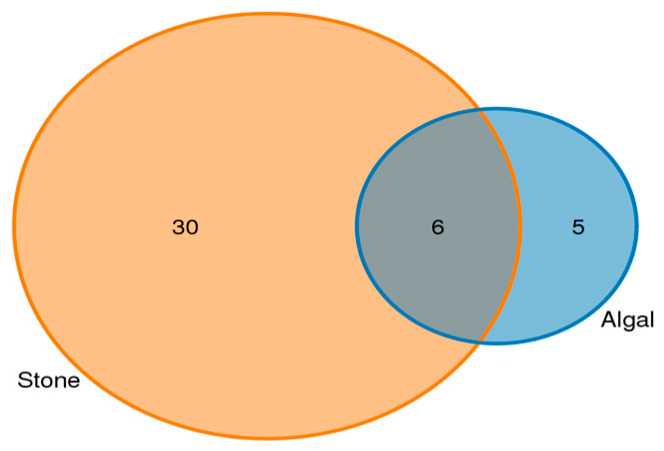
ASVs (amplicon sequence variants) shared by eukaryotic organisms in the gut contents of sea urchins from different regions.

**Figure 8 biology-13-00623-f008:**
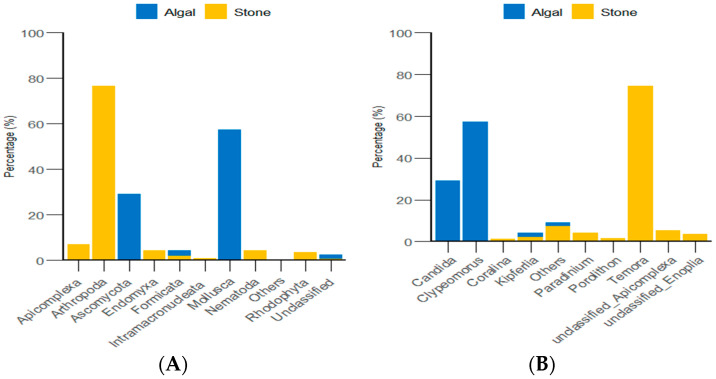
Abundance of eukaryotic organisms in sea urchin gut contents at the phylum level in both areas. (**A**) Proportion of the gut microbiota at the phylum level in sea urchins. (**B**) Proportion of the gut microbiota at the genus level in sea urchins.

**Figure 9 biology-13-00623-f009:**
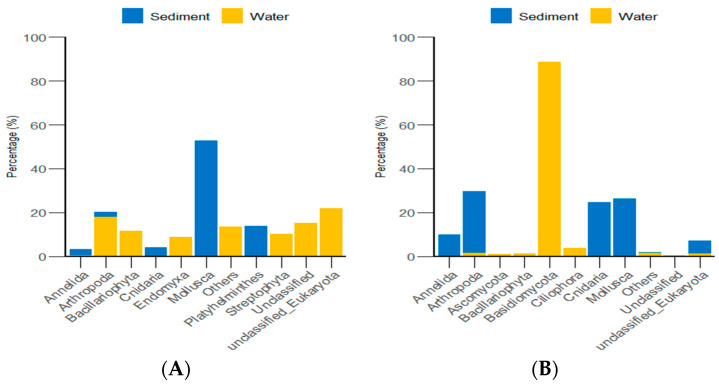
Abundance of eukaryotic organisms in the environment at the phylum level in both areas. (**A**) represents the algal area. (**B**) represents the stone area.

**Figure 10 biology-13-00623-f010:**
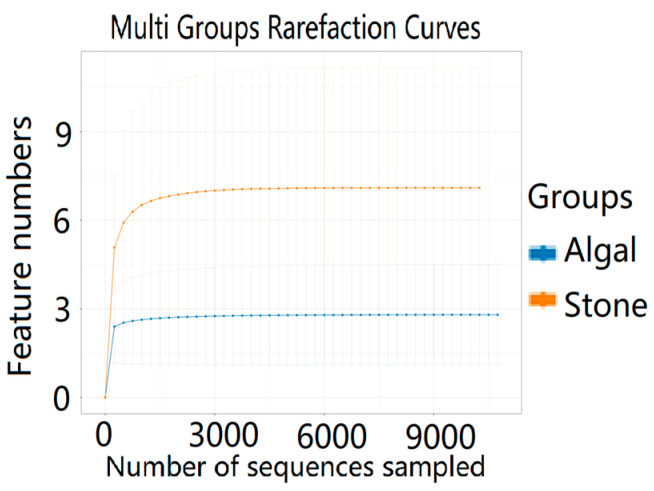
Dilution curves of eukaryotic organisms in the guts of sea urchins from different regions.

**Figure 11 biology-13-00623-f011:**
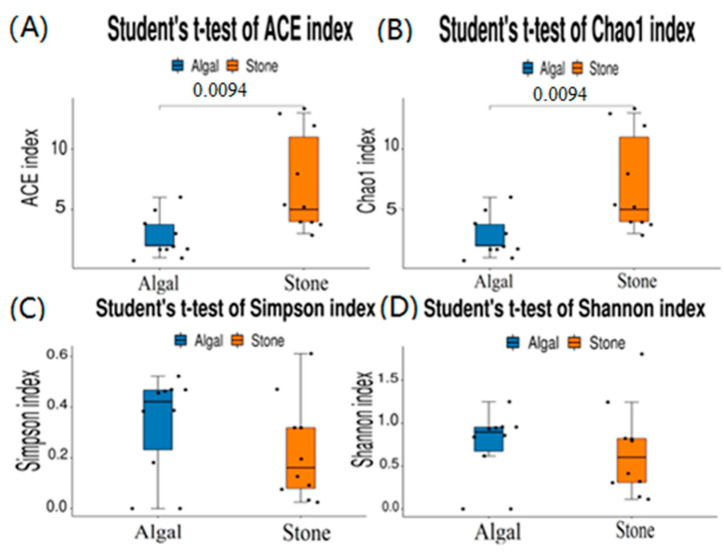
Alpha diversity of eukaryotes in the guts of sea urchins from different regions. (**A**) ACE index analysis of the eukaryotic organisms in the guts of sea urchins from different habitats (*p* < 0.05). (**B**) Chao1 index analysis of the eukaryotic organisms in the guts of sea urchins from different habitats (*p* < 0.05). (**C**) Simpson index analysis of the eukaryotic organisms in the guts of sea urchins from different habitats (*p* > 0.05). (**D**) Shannon index analysis of the eukaryotic organisms in the guts of sea urchins from different habitats (*p* > 0.05).

**Figure 12 biology-13-00623-f012:**
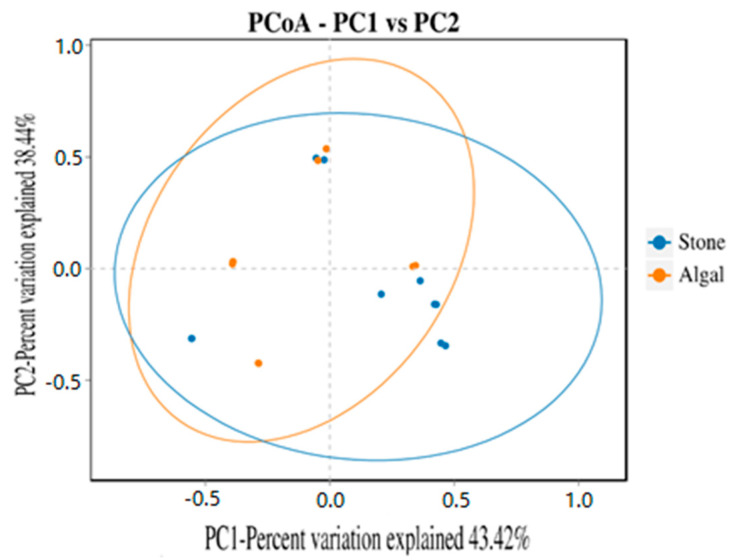
PCoA of sea urchin eukaryotes in different regions.

**Figure 13 biology-13-00623-f013:**
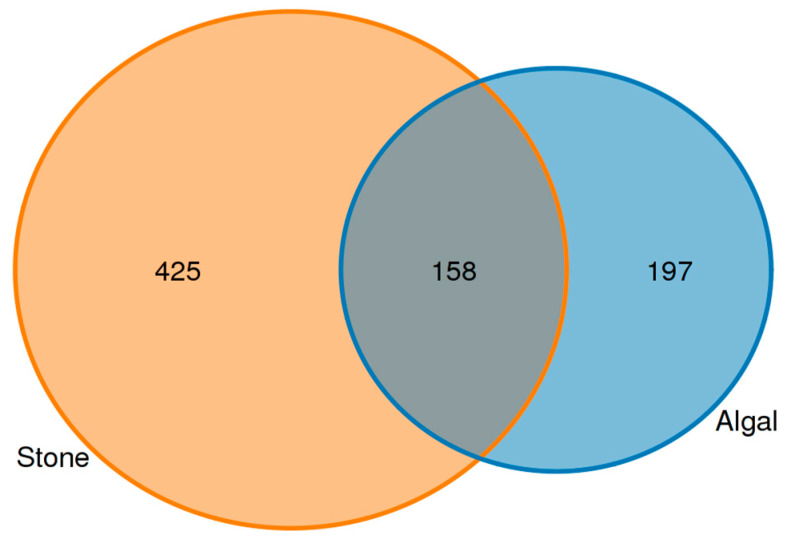
Common ASVs of intestinal microbes in different regions of sea urchins.

**Figure 14 biology-13-00623-f014:**
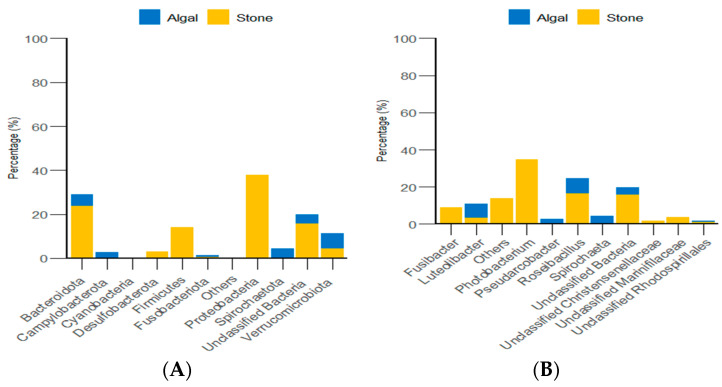
The abundance of gut micro-organisms in the two regions at the phylum and genus level. (**A**) Represents the phylum level. (**B**) Represents the genus level.

**Figure 15 biology-13-00623-f015:**
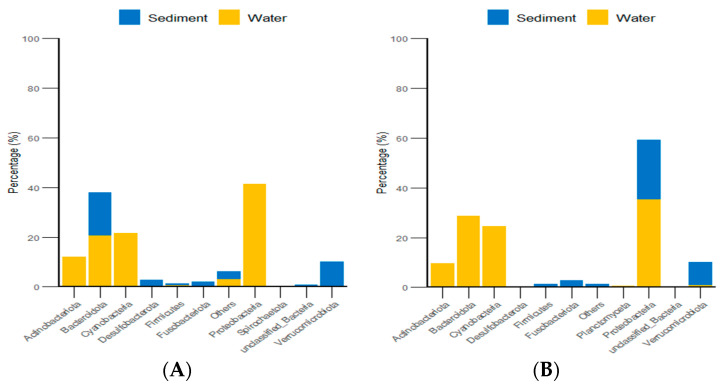
Abundances of environmental micro-organisms at the phylum level in the two zones. (**A**) Represents the algal area. (**B**) Represents the stone area.

**Figure 16 biology-13-00623-f016:**
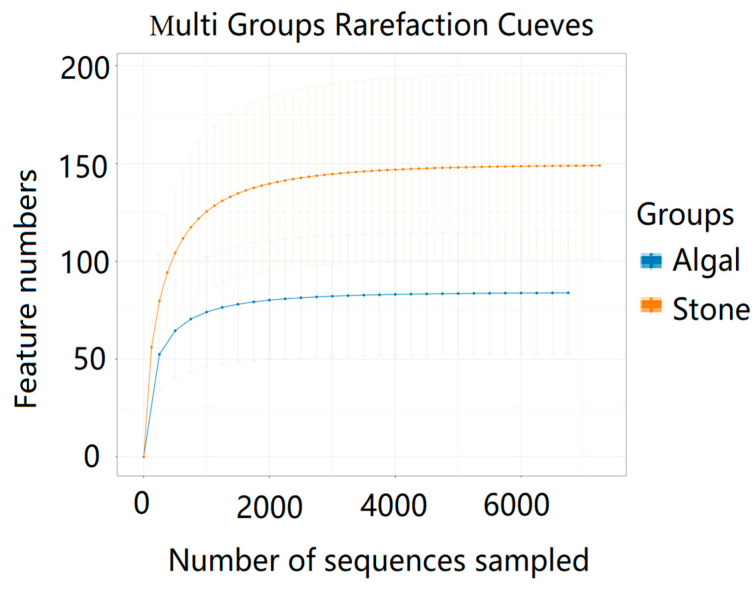
Dilution curves of the gut microbes of sea urchins from different regions.

**Figure 17 biology-13-00623-f017:**
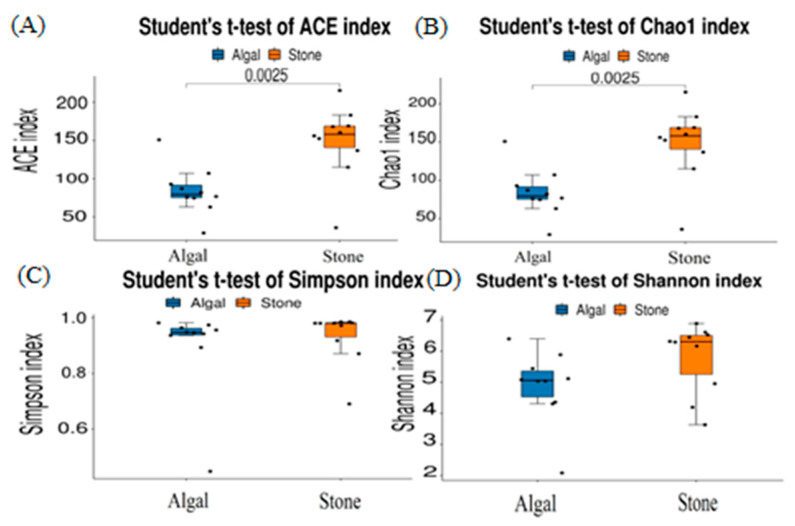
Alpha diversity of the intestinal microbial communities of sea urchins in different regions. (**A**) ACE index analysis of the eukaryotic organisms in the guts of sea urchins from different habitats (*p* < 0.05). (**B**) Chao1 index analysis of the eukaryotic organisms in the guts of sea urchins from different habitats (*p* < 0.05). (**C**) Simpson index analysis of the eukaryotic organisms in the guts of sea urchins from different habitats (*p* > 0.05). (**D**) Shannon index analysis of the eukaryotic organisms in the guts of sea urchins from different habitats (*p* > 0.05).

**Figure 18 biology-13-00623-f018:**
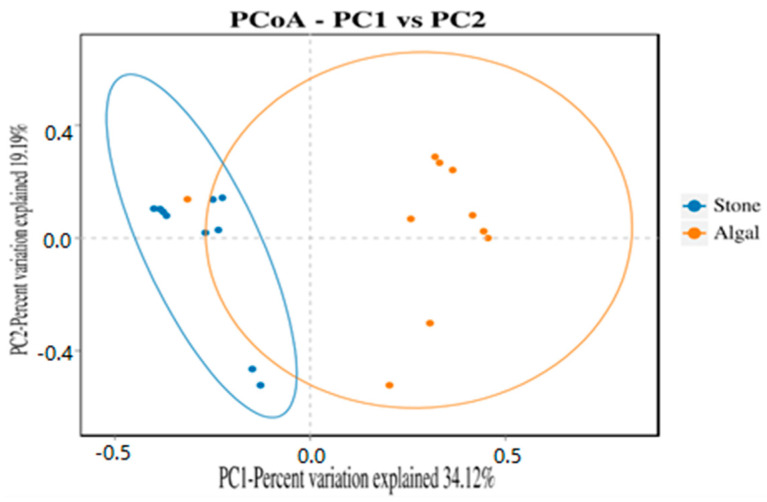
PCoA of the gut microbes of sea urchins in different regions.

**Table 1 biology-13-00623-t001:** Stable isotope values of the purple sea urchin and its potential food sources in different areas.

Alpha Diversity Indices	Stone Region	Algal Region
ACE	2.8 ± 0.53	7.1 ± 1.29
Chao1	2.8 ± 0.53	7.1 ± 1.29
Simpson	0.23 ± 0.06	0.33 ± 0.06
Shannon	0.68 ± 0.17	0.74 ± 0.13

**Table 2 biology-13-00623-t002:** Alpha diversity indices of intestinal eukaryotes in sea urchins from different regions (means ± standard deviations).

Alpha Diversity Indices	Stone Region	Algal Region
ACE	149.1 ± 15.1	84 ± 9.87
Chao1	149.1 ± 15.1	84 ± 9.87
Simpson	0.93 ± 0.03	0.9 ± 0.05
Shannon	5.8 ± 0.36	4.87 ± 0.37

**Table 3 biology-13-00623-t003:** Alpha diversity indices of intestinal micro-organisms in sea urchins from different regions (means ± standard deviations).

Alpha Diversity Indices	Stone Region	Algal Region
ACE	149.1 ± 15.1	84 ± 9.87
Chao1	149.1 ± 15.1	84 ± 9.87
Simpson	0.93 ± 0.03	0.9 ± 0.05
Shannon	5.8 ± 0.36	4.87 ± 0.37

## Data Availability

All 16S rDNA and 18S rDNA raw data have been deposited in the NCBI database under accession codes SUB14575749 and SUB14564666. All of the data are available from the corresponding authors upon reasonable request.

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
