# Peer review of "High-Throughput Sequencing Analysis Revealed a Preference for Animal-Based Food in Purple Sea Urchins"

_biology, 2024, doi:10.3390/biology13080623_

Round 1

Reviewer 1 Report

Comments and Suggestions for Authors

The authors combined stale isotope analysis and high-throughput sequencing technology to compare the diet of purple sea urchins in the stone and algal areas of Daya Bay, China. Generally speaking, the authors generated numerous data to support the main conclusions, and the overall wiring is good. The present version is acceptable for publication.

Minor issues:

1. Extra editing is required. For example, Line 200: it should be “2.5 16S RDNA”; same mistake happened in line 235.

2. Why is there such a difference in dominant species when using different (16S and 18S rDNA) sequencing techniques?

Comments on the Quality of English Language

Extra editing is required. For example, Line 200: it should be “2.5 16S RDNA”; same mistake happened in line 235.

Author Response

Comments 1: [Extra editing is required. For example, Line 200: it should be “2.5 16S RDNA”; same mistake happened in line 235.]

Response 1: Thank you for pointing this out. We agree with this comment. Therefore, we have added a space symbol to the 200-line and 235-line headers. to correct this error in the updated manuscript.

Comments 2: [Why is there such a difference in dominant species when using different (16S and 18S rDNA) sequencing techniques?]

Response 2: Thank you very much for your valuable question. 16S rDNA and 18S rDNA sequencing are two commonly used molecular biology techniques to study the community structure of bacteria and eukaryotes in microbial ecology research. The large difference between the two techniques in the detection of dominant species is that they detect different types of organisms. 16S rDNA sequencing is a common method for the analysis of bacterial and archaeal communities. The 16S is part of the RNA (rRNA) gene for small subunit RNA (rRNA) of bacteria and archaeal ribosomes; it contains multiple highly conserved regions and some variable regions. These variable regions can be used to distinguish between different bacterial and archaeal species. As a result, 16S rDNA sequencing can reveal the bacterial and archaeal composition of a sample. 18S rDNA sequencing is a common method for eukaryotic community analysis. It detects a ribosomal gene found only in eukaryotes. Through 18S rDNA sequencing, researchers can understand the composition of eukaryotic communities in samples, including fungi, protozoa, algae, etc. In summary, the large differences between 16S and 18S rDNA sequencing technologies in terms of dominant species are due to their analysis of bacteria/archaea versus eukaryotes, respectively, which are evolutionarily distinct, and their adaptations and ecological niches in different environments.

2.Response to Comments on the Quality of English Language

Point 1: [Extra editing is required.]

Response 1: We appreciate your guidance and have addressed all of the expert's suggestions accordingly. We are pleased to inform you that we have engaged American Journal Experts for the English editing of our article,Please see the attachment.

Reviewer 2 Report

Comments and Suggestions for Authors

The abstract could have ended briefly describing the future prospectives of this research study. The introduction could have started with the importance of purple sea urchins which has described in the later part of the introduction followed by why is it needed to study with new technology? All the figures and the font size need to be larger. Certain figures like Figure 18 are not readable at all. The legends below the figures could be descriptive too. In the discussion, is there any genetic differences among purple sea urchins that affect their preference for animal-based foods? The conclusion could have been ended with detailed future prospectives or what is the study’s future potential. Any thoughts to include the possible implications for its conservation efforts or management policy in the conclusion? Overall, very interesting manuscript. 

What is the main question addressed by the research? The main aim of this manuscript is to understand the diet and the feeding habits of purple sea urchins using the new technology so that certain restoration strategies could be planned.
Is it relevant and interesting? Yes, this was an interesting manuscript for the reader
How original is the topic? Definitely a lot of studies are going on these sea urchins. This group attempted to find out about certain important characteristics so that they could be conserved in future.
What does it add to the subject area compared with other published
material? This group attempted to find out about certain important characteristics so that they could be conserved in future. Is the paper well written? YES
Is the text clear and easy to read? YES
Are the conclusions consistent with the evidence and arguments presented? Mostly yes but they have certain future potential missing in the end.

Do they address the main question posed? Yes                                                                                                                         

Author Response

Comments 1: [The abstract could have ended briefly describing the future prospectives of this research study.]

Response 1: Thank you for pointing this out. We agree with this comment. In this context, we have revised the end of the abstract to add future prospects for this research study. Please see lines 31, 32, and 33 of the updated manuscript.

Comments 2: [The introduction could have started with the importance of purple sea urchins which has described in the later part of the introduction followed by why is it needed to study with new technology? ]

Response 2: Thank you for pointing this out. We agree with this comment. We have changed the order of the narration in the introduction section of the manuscript, which you can view in the revised manuscript. It also involves changing the order of the references, which we have also adjusted.

Comments 3: [All the figures and the font size need to be larger. Certain figures like Figure 18 are not readable at all. The legends below the figures could be descriptive too.]

Response 3: Thank you for pointing this out. We agree with this comment. All images in this manuscript have been enlarged in font so that you can see them clearly. At the same time, we have also added a legend to the diagram so that you can better understand it in the new manuscript.

Comments 4: [In the discussion, is there any genetic differences among purple sea urchins that affect their preference for animal-based foods?]

Response 4: Thank you very much for your valuable question. We agree with your comment. Genetic differences among sea urchins may also be a major reason for sea urchins' preferences for different animal-based foods. However, since we did not explore this in the current study, we will need to determine this in our future work.

Comments 5: [The conclusion could have been ended with detailed future prospectives or what is the study’s future potential. Any thoughts to include the possible implications for its conservation efforts or management policy in the conclusion?]

Response 5: Thank you very much for your suggestion. We would like to agree with your comments, and we conclude by adding the potential for future development of this study in the conclusion section. “The study of sea urchin feeding habits is highly valuable for marine ecology, resource management, biodiversity conservation and sustainable development of the marine economy. This study contributes to the development of effective conservation measures and management policies that promote the sustainable use of marine resources and the health of ecosystems. In the future, this research is expected to make significant contributions to the sustainable development of human society and the conservation of marine resources.”

Reviewer 3 Report

Comments and Suggestions for Authors

The article titled "High-Throughput Sequencing Analysis Revealed a Preference for Two Animal-Based Foods in Purple Sea Urchins" is an in-depth examination of the dietary preferences of purple sea urchins (Strongylocentrotus purpuratus) using advanced sequencing technologies. This study aims to elucidate the specific animal-based food sources that purple sea urchins favor, contributing to our understanding of their ecological role and dietary behaviors.

Line 41 eliminates “and” before constructing, and changes “provide” by providing

Line 101 Change “Our aim was” by “We aimed”

Line 111: Eliminates the word “Literature” and indicate a cite for the sentence: “The potential food sources for S. purpuratus generally include plankton, macroalgae, sediment, and attached diatoms”

Line 114: If you know could indicate the species of Padina or insert Padina sp. / spp.

Line 122: In situ in cursive

Lines 126 and 150: Change the volumes unit ml by “mL”

Line 264 double (,)

Comments on the Quality of English Language

The article titled "High-Throughput Sequencing Analysis Revealed a Preference for Two Animal-Based Foods in Purple Sea Urchins" is an in-depth examination of the dietary preferences of purple sea urchins (Strongylocentrotus purpuratus) using advanced sequencing technologies. This study aims to elucidate the specific animal-based food sources that purple sea urchins favor, contributing to our understanding of their ecological role and dietary behaviors.

This is an excellent and well-written article. I recommend accepting the article for publication with minor revisions to address the discussion of ecological impacts, comparative analysis, and suggestions for future research.

Author Response

Comments 1: [Line 41 eliminates “and” before constructing, and changes “provide” by providing]

Response 1: Thank you for pointing this out. We agree with this comment. Since the introduction has changed in response to another reviewer’s comment, you can view this change in line 69.

Comments 2: [Line 101 Change “Our aim was” by “We aimed”]

Response 2: Thank you for pointing this out. We agree with this comment. This error has been corrected, and you can see it in line 101 of the new manuscript.

Comments 3: [Line 111: Eliminates the word “Literature” and indicate a cite for the sentence: “The potential food sources for S. purpuratus generally include plankton, macroalgae, sediment, and attached diatoms”]

Response 3: Thank you for pointing this out. We agree with this comment. We have corrected the error and added a citation, which you can view in the new manuscript on line 114.

Comments 4: [Line 114: If you know could indicate the species of Padina or insert Padina sp. / spp.]

Response 4: Thank you for pointing this out. We agree with this comment. We have inserted Padina sp./spp. in the text, which you can view in line 116 of the new manuscript.

Comments 5: [Line 122: In situ in cursive]

Response 5: Thank you for pointing this out. We agree with this comment. We have fixed this error in the text, which you can view on line 124 of the new manuscript.

Comments 6: [Lines 126 and 150: Change the volumes unit ml by “mL”]

Response 6: Thank you for pointing this out. We agree with this comment. We have fixed this error in the text, which you can view in lines 128 and 152 of the new manuscript.

Comments 7: [Line 264 double (,)]

Response 7: Thank you for pointing this out. We agree with this comment. We have removed the (,) from the text, and you can view it in line 266 of the new manuscript.

Response to Comments on the Quality of English Language

Point 1: [Extensive editing of English language required]

Response 1: We appreciate your guidance and have addressed all of the expert's suggestions accordingly. We are pleased to inform you that we have engaged American Journal Experts for the English editing of our article, Please see the attachment.
